# Insights into the molecular triggers of parosmia based on gas chromatography olfactometry

Jane K. Parker [1✉], Christine E. Kelly [1,2] & Simon B. Gane [3✉]

## Abstract

**Background** Parosmia is a debilitating condition in which familiar smells become distorted and disgusting, with consequences for diet and mental health. It is a feature of post-infectious olfactory loss, particularly resulting from COVID-19. There is currently little understanding of its pathophysiology, and the prevailing hypothesis for the underlying mechanism is aberrant growth of regenerating olfactory sensory neurons after damage.

**Methods** We use gas-chromatograph olfactometry to individually present components of a complex olfactory mixture as a rapid screening tool for assessment of both quantitative and qualitative olfactory dysfunction in those with and without parosmia. This allows them to report the associated sensory effects and to identify those molecules which are altered or parosmic in nature.

**Results** Here we show 15 different molecular triggers of this symptom. These trigger molecules are common to many in the parosmic volunteer group and share certain characteristics such as extremely low olfactory threshold and common molecular structure

**Conclusions** We posit that specific highly odour-active molecules are the cause of the parosmic symptom in most cases and initiate the sense of disgust, suggesting that parosmia is, at least in part, a receptor-level phenomenon.

**Plain language summary**

During the recovery from smell loss, caused by infection or injury, sometimes certain smells can become revolting – a condition called parosmia. We used a technique that separates out the chemicals that make up the smell of instant coffee and let several people with parosmia after infection smell them one at a time. Most of these people picked out the same chemicals as smelling disgusting and setting off their parosmia. These chemicals are known to have strong smells to humans and can be grouped into four classes based on their chemical shape and the elements they contain. These findings help in the understanding of what chemical compounds trigger parosmia, which may help in developing diagnostics and therapies for this condition in the future.

[1] School of Chemistry, Food and Pharmacy, University of Reading, Reading, UK. [2] AbScent, 14 London Road, Andover, Hampshire, UK. [3] Royal National Ear, Nose and Throat and Eastman Dental Hospitals, University College London Hospital, 47-49 Huntley St, London, UK. ✉email: j.k.parker@reading.ac.uk; simongane@nhs.net

Prior to the COVID-19 pandemic, olfactory dysfunction was largely unrecognised, and often underestimated by health care professionals. Since the spread of SARS-CoV-2, and the realisation that 50–65% of cases result in anosmia[1] (the loss of sense of smell), there is a greater awareness of the debilitating effect of olfactory disorders[2]. Typically, in cases post COVID-19, normal olfactory function returns within a few weeks, but one study estimates 12% of all cases result in long term smell dysfunction[3]. With >200 million confirmed cases of COVID-19 worldwide[4], this is a significant problem facing the global population today.

Parosmia often occurs in the early stages of recovery from anosmia, typically 2–3 months after onset[1], particularly in those whose anosmia was either acquired post-infection. It is characterised by episodes of triggered olfactory distortions in which familiar everyday smells become altered and unpleasant, to the extent that they become almost unrecognisable, and these distortions vary in strength and duration[5]. Note that throughout the paper, when we refer to parosmia triggers, we are referring to triggers of the episodes rather than triggers of disease onset. Those severely affected find their quality of life deteriorates as everyday activities such as eating, showering and social interactions become a challenge. They report being distressed and anxious about their future[5] and, with many food aromas being intolerable, they start to reject food, leading to significant changes in weight[6], a decline in mental health and, in severe cases, to clinical depression[7,8]. Although many mechanisms for parosmia have been proposed, there is very little fundamental understanding of its pathophysiology.

The aim of this work was to gain insight into the mechanisms involved in parosmia. In 2013, coffee and chocolate were found to elicit distorted olfactory experiences in parosmia[9] and more recently, coffee, meat, onion, garlic, egg, mint/toothpaste were identified in a thematic analysis of group posts on social media[5]. These foods contain aroma compounds with some of the lowest odour-thresholds known, and we suggest that these compounds may be involved in triggering episodes of parosmia. Our original hypothesis was based on that of Leopold[10] who proposed that parosmia was a result of incomplete characterisation of the odorant. As olfactory sensory neurons (OSN) regenerate from basal stem cells, selective detection of just the pungent highly odour-active compounds might result in an incomplete, and therefore distorted perception of certain foods and beverages. Whether this would be sufficient to cause the strong sense of disgust, often reported with parosmia, was not clear.

Our approach is to use GC-Olfactometry (GC-O) to determine which of the aroma compounds present in the headspace of coffee are responsible for distortions and the sense of disgust experienced by those with parosmia. Gas chromatography separates the hundreds of volatile components present in the sample headspace which, when coupled to an odour-port, allows subjects to sniff and describe each component as it elutes from the column and assess a variety of single aroma compounds in a short time.

In this paper we demonstrate that there are a small number of highly potent odorants responsible for the parosmia stimulus when smelt by those with parosmia. These odorants fall into four groups based on their physio-chemical characteristics which implies that only a small number of olfactory receptors are responsible for the sensation.

## Methods

**Participants**. This study (No 22/19) was approved by the University of Reading Research Ethics Committee. All participants received full information and gave their informed consent. All parosmic participants were recruited via Facebook support groups or local ENT consultants. The major inclusion criterion was those with post-infection olfactory loss (the aetiology most likely to result in parosmia[11]), whilst those with other aetiologies such as degenerative olfactory loss or traumatic brain injury were excluded from this study. Non-parosmic participants were recruited from within the Department of Food and Nutritional Sciences at the University of Reading, or through private Facebook pages. The initial study was carried out with pre-COVID-19 parosmic participants ($N = 14$) and non-parosmic participants ($N = 15$) between October 2019 and March 2020. This was supplemented with post-COVID-19 parosmic participants ($N = 15$) between July and September 2020. All volunteers completed a screening questionnaire (see Supplementary Information) before attending a study day in the Olfaction Laboratory at the University of Reading. Selection was based on the participants listing coffee as a key trigger, and answering "often" at least once to two key questions which have been reported to discriminate most efficiently between parosmic participants and those with quantitative olfactory disorders[12]:

i. Are odours that are pleasant to others, unpleasant to you? Never/rarely/often/always
ii. Is the taste of food different to what you expect? Never/rarely/often/always

**Olfactory function**. The bilateral olfactory function of all participants was assessed at the beginning of the day using the well-established and validated orthonasal psychophysical Sniffin' Sticks test (Burghart, Wedel, Germany)[13], based on the threshold of 2-phenylethanol (T), discrimination (D) and identification (I) tests. The resulting TDI scores, which range from 0 to 48, gives a measure of quantitative olfactory function.

**Rationale for use of coffee**. A cocktail of individual aroma compounds was initially considered for the GC-O study, but mixtures of compounds, especially those containing sulfur, are unstable, prone to oxidation, interact with each other and the solvent, and are onerous to prepare from fresh for each subject. This approach is also dependant on pre-selection of the likely trigger molecules from a base of thousands of volatile compounds, all of them potential triggers. The use of a foodstuff allowed screening of a range of volatile compounds, both triggers and non-triggers, in a more stable environment. Coffee was selected as it has been recognised on several occasions to be a major trigger of parosmia[5] and has the additional advantage of being widely consumed. However, coffee aroma is highly variable, and degrades over time. Our solution was to use catering sachets of pre-portioned, one mug instant coffee, to produce a material as consistent as possible, which would last the duration of the study. In effect, instant coffee was being used as a stable carrier for a wide range of potential trigger and non-trigger molecules.

**Gas chromatography-mass spectrometry (GC-MS)**. For standard coffee analysis, each sachet was made up of 300 ml of boiling water and the headspace extracted using solid phase microextraction (SPME). A concentrated sample prepared in 3 ml of water was also prepared to aid the identification of the aromas detected. Both standard and concentrated extracts were analysed by GC-MS using a typical program on a non-polar column and also on a polar column to confirm compound identities. Full details of the procedure are provided in the Supplementary Methods.

**Gas chromatography-olfactometry (GC-O)**. All volunteers assessed the standard coffee extract using GC-O on a non-polar

column. In addition, three parosmic volunteers and two experts also assessed the coffee extract on a polar column to confirm compound identities. Full details of the chromatography are provided in the Supplementary Methods.

**Procedure at the odour-port**. Subjects were sat in front of the GC-O with their nose placed in, but not resting on, a glass cone. They were familiarised with the instrument, instructed to breathe normally during the run, and advised that they could stop at any time. As the aromas eluted from the column, three bits of information were requested from the subjects: (i) an odour description, (ii) an odour intensity, and (iii) an indication of whether the odour elicited a parosmic response. Since the description and identification of aromas in the absence of any other cues is difficult, all participants were presented with a fla-vour wheel before they started (Supplementary Fig. 1), which they could use as a reference during the GC-O run. It had been developed by two experts who sniffed samples of the same coffee (both at regular strength and concentrated) by GC-O. The words were categorised into food and non-food, and colour coded for quick reference. The flavour wheel was of more use to non-parosmic participants, as parosmic participants found it hard to describe many of the aromas, even with the help of the flavour wheel. Many resorted to using the terms "new coffee", "that parosmia smell", "trigger number 1" or "trigger number 2". As each aroma eluted, parosmic participants were prompted to highlight anything that had a parosmic character or trigger. Intensity was scored on a general labelled magnitude scale (gLMS) with anchors at barely detectable, weak, medium, strong, very strong and strongest imaginable. This was chosen over the more common visual analogue scale to allow for instances where parosmic participants wanted to extend the range of scores upwards. All subjects carried out the GC-O of coffee twice. During the second run, the focus was on refining the descriptors with discussion between the researcher and the subject to help identify the compounds eluting.

**Confirmation of identity of the trigger molecules**. Supplementary Data 1 shows how the identification of each trigger was confirmed based on comparison of the mass spectrum, linear retention index and odour character with those of authentic standards. Three parosmic participants returned to assess coffee on a polar column to confirm the identity of trigger compounds. Once identified, selected trigger compounds diluted in mineral oil or propylene glycol at 10 mg L$^{-1}$ were presented to two parosmic participants as described for the European test of olfactory capabilities[14]. They were asked to sniff the vial and indicate whether each compound released "that parosmia smell" which they had described previously.

**Additional samples**. Extracts of cocoa, meat, peanut butter, and red pepper were prepared for coffee with modifications described in Supplementary Methods. Human faecal samples were obtained with informed consent and kindly prepared under Class 2 conditions by members of the Food and Microbial Science Unit at the University of Reading with ethical approval from Reading Research Ethics Committee (number UREC 1520). The sample was mixed with an equal weight of water, and 3 g transferred to an SPME vial. Chromatography conditions were the same as for coffee.

**Reporting summary**. Further information on research design is available in the Nature Research Reporting Summary linked to this article.

## Results

**Participants**. Table 1 shows demographic data for all participants ($N = 44$) which includes 29 participants reporting post-viral parosmia (PAR) and 15 without parosmia (NONPAR) (see Supplementary Data 2). All participants were non-smokers and self-reported that they could taste the difference between salt and sugar. Pre-COVID-19 PAR and NONPAR were age-matched with mean ages of 56 and 49 y respectively and there was no significant difference in age between the two groups ($p = 0.12$). Post-COVID-19 PAR were significantly younger (mean age 37 y) than their pre-COVID-19 counterparts and the NONPAR group ($p < 0.000$, $p = 0.008$ respectively).

**Olfactory function**. TDI scores, obtained from the Sniffin' Sticks test, were significantly lower in pre- and post-COVID-19 parti-cipants (mean 28 and 27 respectively) compared to the NONPAR group (mean 37) (ANOVA, $p < 0.0001$ respectively) but there was no significant difference between pre- and post-COVID-19 groups ($p = 0.71$). The TDI scores of the combined parosmic groups ranged from functionally anosmic to normosmic (10–38). Ten of this group were classified as normosmic on raw TDI score, increasing to 17 (more than half the group) when age adjustment was applied[15], whereas two scored <16 and were classified as functionally anosmic, even though they were able to perceive some triggers of parosmia. We demonstrate here that although on average most parosmic participants had a low olfactory function, parosmia also occurs in those with a normal olfactory function and those who are functionally anosmic.

**Gas chromatography-olfactometry**. PAR (pre- or post-COVID-19) detected significantly fewer aromas at the GC-odour port than NONPAR ($p < 0.0001$) (means 20, 19 and 37 respectively). The number of GC-O aromas correlated well with the TDI score ($R^2 = 0.66$, Fig. 1a) as both are indicators of quantitative olfactory function. However, the mean number of aromas which triggered parosmia in PAR was only 6 (range 0–13) indicating that on average they detected three times more "normal" aroma mole-cules than trigger molecules—one of our most important findings that demonstrates that not all aroma compounds are triggers (Supplementary Data 1). The role of individual molecules in triggering parosmia has never been demonstrated before and this work suggests that, in those presenting with parosmia, it is spe-cific molecules which trigger the altered perception of food and the sense of disgust. There was no strong correlation between the

**Table 1 Summary of participant demographics.**

| Participant demographic data | No | Male | Female | Age (mean) | Age (range) | Age (SD) | CRS |
|---|---|---|---|---|---|---|---|
| All Participants | 44 | 12 | 32 | 47 | 19–73 | 14 | 4 |
| Pre-COVID-19 parosmics | 14 | 3 | 11 | 56 | 33–73 | 9.6 | 1 |
| Post-COVID-19 parosmics | 15 | 3 | 12 | 37 | 19–60 | 12.2 | 1 |
| Non-parosmics | 15 | 6 | 9 | 49 | 33–71 | 13.2 | 2 |

CRS chronic rhinosinusitis.

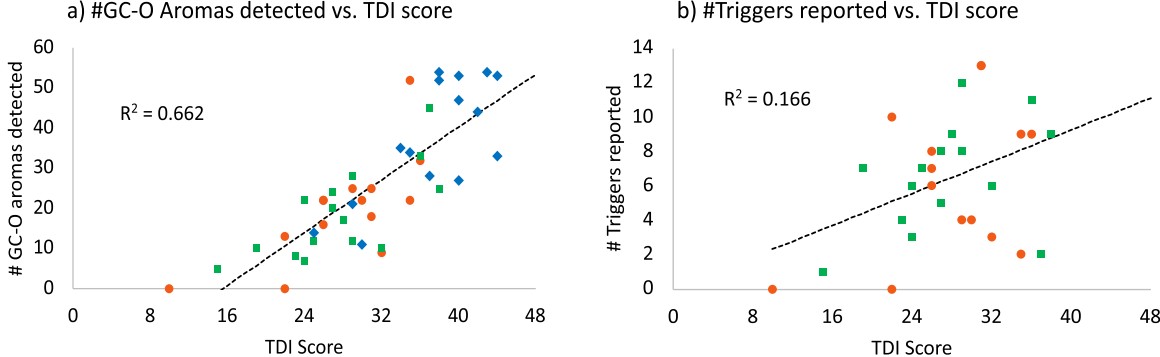

**Fig. 1 Correlations between olfactory function and GC-O. a** Correlation between TDI score (Threshold, Discrimination, and Identification Sniffin' Sticks test score) and number of aromas detected at the GC odour-port. **b** Relationship between number of triggers detected in the coffee extract and TDI score. In both figures, non-parosmic participants = blue, pre-COVID-19 parosmic participants = orange, post-COVID-19 parosmic participants = green. Source data in Supplementary Data 2.

**Table 2 Compounds most frequently detected by parosmic participants.**

| Molecular triggers | Code | Odour threshold ug/L | Number times detected by parosmic | Number times reported as trigger |
|---|---|---|---|---|
| 2-furanmethanethiol | **T1** | 0.005[16] | 24 | 20 |
| 2-ethyl-3,6-dimethylpyrazine | **P1** | 0.01 | 18 | 15 |
| 2,3-diethyl-5-methylpyrazine | **P2** | 0.05[16] | 20 | 13 |
| 2-furanmethyl methyl disulfide | **D1** | 0.04[20] | 18 | 11 |
| 2-methyl-3-furanthiol | **T2** | 0.0004[18] | 19 | 10 |
| 2-methyl-3-furyl methyl disulfide | **D2** | 0.004[18] | 18 | 10 |
| 2-ethyl-3,5-dimethylpyrazine | **P3** | 1[37] | 17 | 10 |
| 3-methyl-2-butene-1-thiol | **T3** | 0.01[19] | 21 | 9 |
| 2-ethyl-3-methoxypyrazine | **M1** | 0.4[21] | 12 | 9 |
| 2-isobutyl-3-methoxypyrazine | **M2** | 0.002[16] | 17 | 7 |
| 3-mercapto-3-methylbutanol | **T4** |  | 13 | 6 |
| 3-hydroxy-4,5-dimethylfuran-2(5H)-one (sotolone) | **X1** | 0.5[16] | 10 | 6 |
| 3-mercapto-3-methylbutyl formate | **T5** |  | 15 | 5 |
| 2-methoxyphenol (guaiacol) | **X2** | 12[16] | 14 | 5 |
| trimethylpyrazine | **P4** | 9.6[21] | 10 | 5 |
| unknown LRI 981 | **X3** |  | 12 | 4 |
| 2-isopropyl-3-methoxyprazine | **M3** | 0.001[16] | 15 | 3 |
| 2,3-butanedione | **X4** | 1[16] | 15 | 2 |
| 4-ethylguaiacol | **NT1** | 4[16] | 10 | 0 |
| (E)-β-damascenone | **NT2** | 1[16] | 9 | 0 |

*T thiol, P pyrazine (trisubstituted), D disulfide, M methoxypyrazine, X unclassified, NT non-trigger.*

number of molecular triggers reported and TDI score ($R^2 = 0.16$, Fig. 1b) suggesting that although quantitative and qualitative olfactory disorders may occur together, their mechanism may be quite different.

**Molecular triggers**. Over 30 different molecules were detected by PAR as a group. The 20 most frequently detected are shown in Table 2 (see Fig. 2 for structures). Of these, 18 were reported to trigger the sense of distortion. The most frequently reported trigger is 2-furanmethanethiol (**T1**) which has an exceptionally low odour threshold in water ($0.004\,\mu g\,kg^{-1}$ [16]). Whereas NONPAR used a range of food-related terms to describe it (coffee, roasty, popcorn, smoky), PAR often struggled to find suitable descriptors, as they were unable to relate it to anything they had smelled before. PAR typically used words describing its hedonic quality (disgusting, repulsive, and dirty) or new coffee (relating to the altered smell of coffee since onset of parosmia) as described previously[17]. Four PAR described it in the same way as NONPAR (biscuit, toasty or roasty) indicating that it is not

universally parosmic, but certainly an important and frequent molecular trigger of parosmia. All NONPAR except one detected this compound.

The equally potent 2-methyl-3-furanthiol (**T2**) (threshold $0.0004\,\mu g\,kg^{-1}$ in water[18]) and its corresponding methyl disulfide (**D2**) were also detected but reported less frequently as distorted. They are character impact compounds in meat, and we confirmed in four parosmic participants who assessed grilled chicken by GC-O that these compounds also triggered parosmic responses to meat.

2-Ethyl-3,6-dimethylpyrazine (**P1**) was the second most frequent trigger in coffee, described with a variety of food terms by NONPAR, but by "new coffee", "unpleasant" and "distorted" by PAR. Some could distinguish it from **T1**, but others could not. Other trisubstituted pyrazines (2,3-diethyl-5-methylpyrazine (**P2**), 2-ethyl-3,5-dimethylpyrazine (**P3**) and trimethylpyrazine (**P4**)) were common triggers. These pyrazines are highly odour-active compounds in roasted, fried and baked goods, and we confirmed by GC-O that these compounds also triggered a parosmic

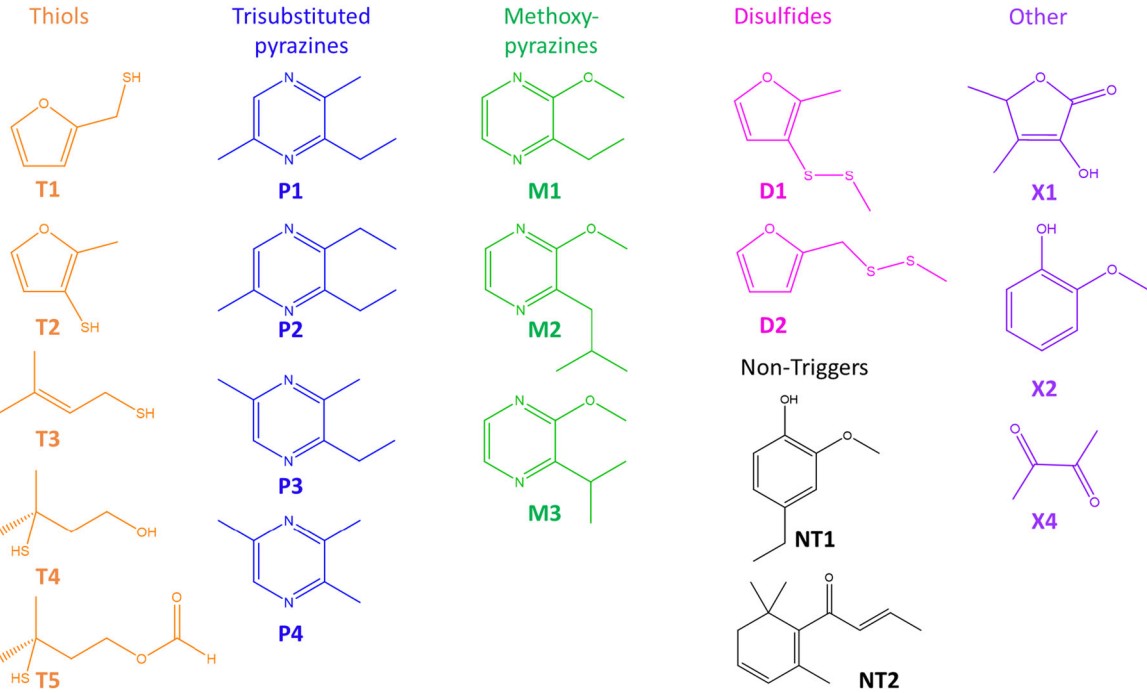

**Fig. 2 Structures of the most frequently detected compounds.** The most common trigger molecules are grouped into four distinct categories based on structure: thiols, trisubstituted pyrazines, methoxypyrazines, and disulfides; although some less common triggers did not fall into any one of these categories.

response to cocoa ($N = 4$), grilled chicken ($N = 4$), and peanut butter [$N = 3$]. 2-Ethyl-3-methoxypyrazine (**M1**), 2-isobutyl-3-methoxypyrazine (**M2**) and 2-isopropyl-3-methoxypyrazine (**M3**) were common triggers in coffee, and we confirmed that these also contributed to the parosmic character of bell peppers ($N = 5$), where they are character impact compounds.

Another thiol, 3-methyl-2-butene-1-thiol (**T3**), with a pungent weedy character and low threshold (0.0002 µg L$^{-1}$ [19]), was reported as a trigger 9/29 times. Although not heterocyclic like the others, it contains the same α,β-unsaturated thiol moiety as **T1**. The polyfunctional thiols, 3-mercapto-3-methylbutanol (**T4**) and its formyl ester (**T5**), are potent aroma compounds in coffee[20] and were detected in half the cases, but only reported as distorted 5 or 6 times. The unknown compound (**X3**) has been tentatively identified as 4-methylthio-4-methyl-pentan-2-one, but this is yet unconfirmed.

Although thiols and disulfides seem to effectively trigger a parosmic response, there are two notable exceptions. Methanethiol (odour threshold 0.02 µg L$^{-1}$ [21]), which was detected by some NONPAR, was not detected by any PAR. Likewise, dimethyl trisulfide (0.01 µg L$^{-1}$ [16]) is an exceptionally potent compound detected by 12/15 NONPAR but only by 4 PAR, and only reported once as a trigger. Although the thresholds are low, there may be insufficient quantities of these compounds present to achieve threshold concentrations for PAR, and further tests are required.

Furthermore, a few compounds were detected but never reported as triggers. 4-Ethylguaiacol (**NT1**) was detected by 7 PAR and always described as spicy, sweet and smoky, but never parosmic. Similarly, (*E*)-β-Damascenone (**NT2**), a key odour-active compounds in coffee with a low odour threshold (0.01 µg kg$^{-1}$ [16]), was detected by 6 PAR and always described as jammy and fruity.

**Principle component analysis**. Principal component analysis was carried out on the intensity data (Supplementary Data 2) for PAR

for the 20 most frequently detected compounds (Fig. 3). The compounds scored with the greatest intensity tended to have a greater component on PC1, whereas PC2 separated the three most frequently detected thiols (**T1, T2, T3**) from the three most frequently detected pyrazines (**P1, P2, P3**). Furthermore, the two disulfides are positioned close to each other (**D1, D2**) and close to their parent thiols, and the two branched methoxypyrazines (**M2, M3**) are also close together. There is some evidence of a structure activity relationship emerging suggesting, for example, that some participants might perceive thiols more intensely and others may perceive pyrazines more intensely.

**Faecal odours**. Volatiles such as skatole and indole are perceived by most people as among the most objectionable odours and are present in faeces[22]. Those suffering from parosmia often comment that the smell of faeces is never as unpleasant as before, often smelling like other distorted foods, or more pleasant and biscuity[5], presenting the interesting corollary that foods smell of faeces yet faeces smell of food. Two parosmic researchers who carried out GC-O on the headspace of a 50% faecal slurry in water did not detect these compounds and were unaware of any foul smells. However, they detected several other compounds, many of which they had also detected in coffee, and only some of which triggered parosmia. In comparison, a normosmic scored the intensity of indole and skatole as close to the strongest imaginable. This provides a neat explanation as to why the changes in valence for faecal samples is reversed. In the absence of signals from the compounds usually associated with disgust in faecal odour, PAR detect other potent volatiles in the sample, normally masked by the stench of the faecal compounds. For some, these other compounds may have a positive valence, for others they may be distorted.

**Correlation between ligand structure and odour receptor (OR)?** Identifying a small number of common molecular triggers for parosmia raised the obvious question of an olfactory receptor

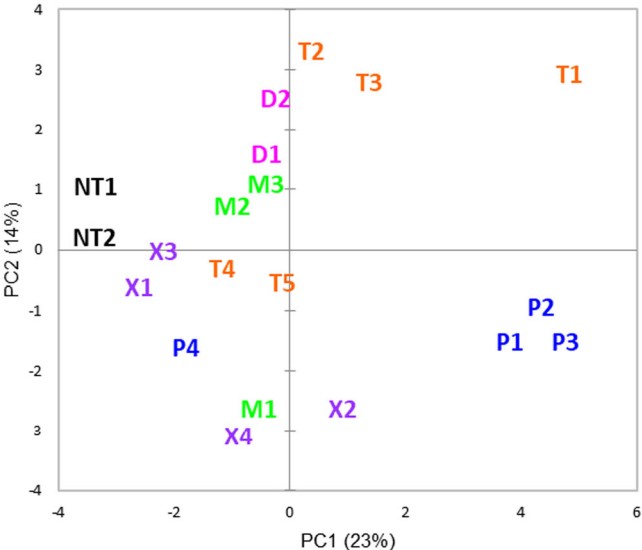

**Fig. 3 Principal component plot (PC1 vs. PC2) for intensity of 20 most frequently detected compounds.** Principal component analysis demonstrating the clustering of the various compounds. Source data are provided in Supplementary Information Table 1.

similarity. To determine whether the clusters are associated with any of the known ligand odour receptor pairs, we searched the ODORactor database[23]. We found no obvious segregation of triggers by olfactory receptor. Most of the triggers activated (with > 50% probability) either OR1G1 or OR52D1. We also compared molecules never reported as triggers such as disubstituted pyrazines, indole, skatole, cresol and found these to activate the same ORs, making it unlikely that these olfactory receptors are the source of the parosmic signal. OR1G1 is known to be very broadly tuned and bind odorants of different chemical classes[24]. Only a fraction of the known ORs have been deorphaned, and further identification of ligand-OR pairs is required to propose any relationship between structure, olfactory receptor, and parosmia.

**Statistics and reproducibility**. The age of the pre-COVID-19 parosmic participants, post-COVID-19 parosmic participants and non-parosmic participants was analysed using Kruskal-Wallis followed by pairwise comparison using Steel-Dwas-Critchlow-Fligner (significance set at 0.05) to determine significant differences between the groups, whereas Anova followed by Tukey HSD ($p = 0.05$) was used for TDI scores. Principal component analysis was carried out on intensity data. All statistical analyses were carried out using XLSTAT version 20201.1.1 statistical and data analysis solution (Addinsoft 2020).

Sample sizes: pre-COVID-19 parosmic participants ($N = 14$), non-parosmic participants ($N = 15$), post-COVID-19 parosmic participants ($N = 15$).

## Discussion
In summary, we have identified for the first time, specific molecules which trigger parosmia. We demonstrate that there is a common set of molecular triggers causing the perception of distortions and a sense of disgust in coffee, and they also trigger distorted perceptions of other chemically related foods. However, not all molecules in this set are triggers for all PAR. These molecules tend to be potent, have very low olfactory detection thresholds and, in isolation, are neither distorted nor unpleasant

for NONPAR. However, odour activity is not the defining factor since (E)-β-damascenone, which has an exceptionally low odour threshold[16], was always perceived as jammy and fruity by both PAR and NONPAR. Most of the trigger molecules found in coffee belong to one of four distinct groups: thiols, pyrazines, disulfides, methoxypyrazines but there are no known odour receptors which are specific for the described trigger molecules. Undoubtedly, there are additional triggers in other foods, beverages, care products and even in the environment, but this subset found in coffee is sufficient to prove our hypothesis.

We have demonstrated that our original hypothesis based on incomplete odorant characterisation (a lack of contribution from other more desirable and less potent aroma compounds[10]) is partly true, in that the highly-odour-active compounds are indeed those that tend to be perceived by PAR, and many other odorants normally perceived by NONPAR are missing. However, what we show goes further by way of an explanation. We have shown that a group of specific highly odour-active compounds are common triggers of distortion and individually elicit the perception of disgust, regardless of how many of the other aroma compounds are perceived at the same time. Some of the PAR perceive as wide a range of odorants as NONPAR, yet the distortions still dominate. Thus, we suggest that the main driver of parosmic episodes is the distorted perception of specific highly odour-active molecules.

Parosmia is a triggered, short-lived, altered smell sensation which almost universally elicits the basic emotion of disgust, but little is known of its pathophysiology. Our finding that the sense of distortion is reliably triggered by a common group of low threshold odorants, advances our understanding of this debilitating condition and places constraints on the prevailing pathophysiological hypotheses. Several mechanisms for parosmia have been proposed[10] and can broadly be thought of as the central theory, the ephaptic theory[25] and the peripheral mis-wiring theory, recent findings not withstanding[26].

The central theory is based on the changes occurring in the integrative centres in the brain. A decrease in olfactory bulb volume[27,28] and a significant loss of grey matter volume has been demonstrated in parosmic patients[29]. Further evidence of a central mechanism shows different fMRI activation patterns in parosmic patients compared to those with hyposmia[30]. Increased activation in the thalamus and the putamen was observed in the parosmic patients, the latter being of relevance since it is connected to the olfactory networks and has been associated with the perception of disgust. Also, stronger activation was observed in the ventral striatum which is associated with odour valence. Whilst there is good evidence in humans for the central theory of parosmia, purely central causation seems unlikely based on our evidence that parosmia is triggered by a group of highly specific molecules at the periphery.

The peripheral mis-wiring theory proposes random mistargeting of OSN to the glomerulus during regeneration. This has been observed in mice with impaired olfactory function[31–35] but not yet in humans, however, it has been adopted as the likely mechanism for parosmia. It is further suggested that the change in hedonic valence is due to broad activation of the olfactory bulb sending a disordered and unmoderated array of signals to the central neural processing system which invokes a strong sense of disgust. Our data neither support nor refute the mis-wiring hypothesis, but certainly place constraints on it, based on the non-random nature of the trigger molecules.

Any proposed mechanism should explain five characteristics: (i) that parosmia arises after widespread destruction of olfactory neurons, either post infection or post traumatic brain injury, (ii) it is triggered by one of several common odorants, (iii) it is of novel odour character, (iv) this character is almost always unpleasant

and that (v) the severity and duration of parosmia can fluctuate quite significantly on a daily basis[5]. Whereas the mis-wiring theory is consistent with the first, and the central theory may explain the novel odour character and the change in valence, none of these hypotheses explain yet why only a few potent molecules elicit such a strong parosmic response and why the response can fluctuate daily. The common molecular structures, low odour thresholds and physiochemical grouping of the molecular triggers of parosmia suggest that this is related to peripheral changes in the olfactory epithelium with downstream consequences. This is consistent with the local damage to the epithelium associated with post-infection olfactory loss but of course, does not exclude any contribution from a central mechanism, and indeed it is likely that both mechanisms are involved.

## Conclusion

In this paper we have used modern interdisciplinary science to explore parosmia in human beings and provide some of the first solid evidence to support its arising in the periphery of the olfactory system. Whilst the use of flavour chemistry techniques has led us to a better understanding of the aetiology and pathophysiology of an increasingly relevant syndrome, our findings also have implications for the development of practical diagnostic tools. An understanding of trigger molecules allows bespoke development of objective tests for parosmia, which are much sought after by patients, researchers, and clinicians alike. Specific triggers could form the basis of such tests, providing better ways of measuring parosmia than questionnaires[12] or hedonic evaluations[36]. Furthermore, our work provides a potential tool to explore more scientifically the underlying mechanism of parosmia.

From a patient perspective, an understanding of trigger foods, on a molecular basis, allows us to provide informative and scientifically sound advice around dietary choices and meal planning for those with post-infectious olfactory disorder, and the clinicians, health professionals and families who care for them. This study represents a significant development in the understanding of this increasingly widespread condition and will guide further research and future therapies.

## Data availability

All data are supplied in the supplementary information, except the mass spectroscopy files which are available in the University of Reading depository https://doi.org/10.17864/1947.000350. Source data for Fig. 1 in the manuscript can be found in "Supplementary Data 1" and for Fig. 3 in "Supplementary Information Table 1".

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

## Acknowledgements

We like to acknowledge all who participated in this study, Aidan Kirkwood for assistance with the participants, Peter Jackson for sourcing the faecal sample, and Professor Barry Smith for useful discussion and review of the manuscript.

## Author contribution

J.P. contributed to the conception, acquisition, analysis, data interpretation, manuscript draft, and review; C.K. contributed to the conception, participant management, data acquisition, and review; S.G. contributed to the conception, data interpretation, manuscript draft, and review (S.G.).

## Competing interests

The authors declare no competing interests.
