## [Peer Review File · Communications Medicine]

Reviewers' comments:

Reviewer #1 (Remarks to the Author):

General Comments :

The authors investigate which odor compounds elicit more likely and reliably parosmia. They did that by means of gas chromatography. The results suggest that certain odors or odor groups are more involved in parosmia triggering than others.

Major Comments:

This is a very interesting and innovative work that will have a major impact in the field.

Methodologically the design and conduction of the study is well done and I do not have major issues concerning methodology and results.

However, I think the way it is presented and written now is very scholar, technical and anaemic. I am in the field for over 20 years and this is a milestone finding for further work to come in parosmia research. This paper does by far not explain anything but is the first in its way to explore parosmia by means of modern science instead of questionnaires. Please go for a short, catchy and nice and easy to read paper!!!

It starts with the title that is not very catchy. I would suggest: "Molecular triggers of parosmia (drop the symptom of): novel insights based on gas chromatography analysis. I would not put underlying mechanisms in the title since the data do not support that.

Second: There is a mix of findings (line 81-182) and anecdotic findings (line 184-213) in individuals. This is weird and breaks the fluid reading. If you stick to this individual botanics, put all that into the supplement findings.

Third: The authors loose too much time for enumerating all the potential mechanisms and try to fit their data into an explanation how parosmia are generated. The present data do not explain much about the mechanism. They suggest a peripheral more than central elements involved but no proof is given with these results. Please shorten all that textbook section on parosmia mechanisms and focus on what you found!

The real contribution you did is the following in my eyes. I would go for: Parosmia is frequent; Parosmia is difficult to measure and it is completely misunderstood how, hwy and why it occurs. Many mechanisms are put forward. We found a way to identify odor components that do reliably and in most patients elicit parosmia. This paper lays the potential ground for a) develop measurement tools in clinical routine to assess parosmia other than by questionnaires [1] or hedonic evaluations [2] and b) opens a potential way how to explore more scientifically the underlying mechanism of parosmia (and maybe phantosmia). That is the way I would sell the paper.

Finally, I have the impression that it is a little written in a too apodictic way. Like any neurological symptom, olfactory distortion (similar to phantom pain, tinnitus, etc..) may have several origins and not only one mechanism. Every neurological symptom may be due to central or peripheral pathologies or sometimes a conjunction of both. Please go for a discussion that considers that. In line with that very black and white view it should also be stated that most patients with parosmia report coffee, cigarette smoke and some other odors as solid triggers, but that an enormous variety of parosmia trigger exist (e.g. even vanilla triggered parosmia has been described [3]). I think the discussion merits mentioning that olfactory distortions are very varying. The present findings are encouraging that we may find ways to measure and "catch" the vast majority of parosmia. Parosmia occurrence is not totally independent of olfactory status. Parosmia in anosmia patients is

rare. But indeed, measured normosmia does not exclude parosmia. Please specify.

References:

Please cite the position paper that gives a good overview on clinical smell disorders. It may help reduce over citing non relevant papers [4]. Some other important papers on parosmia would be nice to be included. Reden et al [5] examined the occurrence of parosmia and phantosmia, showing that parosmia is largely related (much more than posttraumatic!!) to postinfectious events. As POID are associated more to purely peripheral events (local damage to the olfactory epithelium, in contrast to posttrauma) this may further help your discussion.

Shorten the manuscript that will appear to those who do not want to read the supplements by 50 % and make it pleasant to read. Then it will be wonderful!

1. Landis, B.N., et al., Evaluating the clinical usefulness of structured questions in parosmia assessment. *Laryngoscope*, 2010. 120(8): p. 1707-13.
2. Liu, D.T., et al., Assessment of odor hedonic perception: the Sniffin' sticks parosmia test (SSParoT). *Sci Rep*, 2020. 10(1): p. 18019.
3. Frasnelli, J., et al., Clinical presentation of qualitative olfactory dysfunction. *Eur Arch Otorhinolaryngol*, 2004. 261(7): p. 411-5.
4. Hummel, T., et al., Position paper on olfactory dysfunction. *Rhinology*, 2017. 54(Supplement 26): p. 1-30.
5. Reden, J., et al., A study on the prognostic significance of qualitative olfactory dysfunction. *Eur Arch Otorhinolaryngol*, 2007. 264(2): p. 139-44.

Minor Comments:

I had to look up "tetrapartite symptom" since I did not understand this word. After extensive internet surfing I am still not smarter. Would it be possible to start your discussion with a word normal people understand?

Reference 27: the journal is wrong: ORL and Related Spec, NOT J ORL HNS. Please correct.

Reviewer #2 (Remarks to the Author):

Besides the well-known anosmia (loss of the ability to smell), the so-called parosmia is another rare disorder of the olfactory system. This disorder was little recognized in medicine, but with the COVID-19 pandemic it was observed that the virus infection caused anosmia in a certain percentage of the patients. Although most patients recovered after the infection, some developed a long-term dysfunction of the olfactory system, i.e. a parosmia. Such patients describe many food aromas as being intolerable and, thus, often normal food is rejected. In consequence they lose weight and may thus develop further clinical consequences.

The aroma of foods is caused by a mixture of several volatile components in the right concentrations, and today many food aromas have already been unraveled on a molecular basis by the application of the so-called sensomics concept. An important analytical tool of this concept is GC/olfactometry or the so-called GC/sniffing, which helps to identify the odor-active components in

a food aroma distillate from the bulk of odorless compounds. In short, a panelist smells any volatile compound at a sniffing-port after separation of the entire volatile fraction by gas chromatography. This way, the aroma attribute of a single compound can easily be evaluated.

Parker and coworkers used the GC/O approach to detect the differences in aroma attributes evaluated by a group of participants with post-viral parosmia vs. those described by a group of participants without parosmia using the volatile fraction of a coffee beverage. By comparing the correct odor attributes given by the healthy subjects (using literature data on earlier identified coffee aroma compounds) to the “wrong” attributes described by the parosmic panelists, differences appeared and such compounds were assigned by the authors as “molecular triggers of symptoms of parosmia”:

I find the idea of using the GC/O approach very innovative, and the results are for sure very interesting in defining food aroma compounds responsible for the rejection of normal food by parosmic patients.

However, I cannot fully agree with the opinion of the authors with respect to the “underlying mechanism”. Also, sentences like “ Furthermore, it provides the basis for development of a practical diagnostic tool and treatment strategies” (abstract, last sentence)are not supported by analytical data. Therefore, I suggest to tone down many sentences and to avoid speculations in the conclusions part.

In addition, I miss systematic sensory experiments with pure reference compounds in defined concentrations, for example in aqueous solution. This way, the effect of the concentration of a compound on the olfactory epithelium, i.e. an olfactory receptor cell can correctly be studied. Also, aroma profiles of the entire coffee beverage should have been performed to elucidate how the parosmic patients evaluate the beverage itself.

Furthermore, to draw the right conclusions on the structure of compounds triggering parosmia, at least a few studies on positional isomers should have been done.

Therefore, as they stand, although quite interesting, the data are a mainly descriptive. Consequently the manuscript should be shortened considerably. In addition, the use of the coffee beverage and the application of GC/olfactometry should be mentioned in the title.

Response to reviewers:

Molecular triggers of symptoms of parosmia and insights into the underlying mechanism

Jane K. Parker, Christine E. Kelly, Simon B. Gane

We thank the reviewers for taking the time to thoroughly review this MS and for their constructive suggestions. We have addressed all the reviewers' comments in blue text below, in particular reducing the size of the manuscript by about one third and addressing the readability of the document. We really appreciate both reviewers' comments that this is interesting and innovative work that will have a major impact in the field, and believe that by restructuring, reducing, and improving readability, this MS is now much improved and suitable for publication.

Reviewer #1 (Remarks to the Author):

General Comments :

The authors investigate which odor compounds elicit more likely and reliably parosmia. They did that by means of gas chromatography. The results suggest that certain odors or odor groups are more involved in parosmia triggering than others.

Major Comments:

This is a very interesting and innovative work that will have a major impact in the field.

Thank you for this very positive comment.

Methodologically the design and conduction of the study is well done and I do not have major issues concerning methodology and results.

Thank you again for this positive comment.

However, I think the way it is presented and written now is very scholar, technical and anaemic. I am in the field for over 20 years and this is a milestone finding for further work to come in parosmia research.

We find no problem with the study being scholarly, after all this is an academic publication, but have removed much of the technical detail to improve the readability. Details of the method have been moved to the supplementary information and method sections replaced with a more lay description, keeping those sections which we believe would be of general interest and removing details of suppliers, chromatography and statistics. The discussion has been reduced from 4 pages to little over 1 page, as we feel that this is where we over-interpreted the results. The technical language in the whole document has been simplified or removed and we hope that the reviewer feels it is now a little less anaemic.

Thank you, we greatly appreciate your comment that this is a milestone finding, and we have included this phrase in the MS at Line 347.

This paper does by far not explain anything but is the first in its way to explore parosmia by means of modern science instead of questionnaires.

We agree – our aim was not to explain parosmia but to discuss current hypotheses in light of our findings. We have removed the bulk of the discussion where we attempted to rationalise (or explain)

our findings. However, we have retained some of the discussion as we believe that any scientific paper should discuss the results in the context of the current literature.

Please go for a short, catchy and nice and easy to read paper!!!

Having viewed papers published recently in COMMSMED, the style is not generally “short and catchy” but contains comprehensive technical detail with which to justify the findings. With our novel approach and new findings, we feel that much (but indeed not all) of what we have written needs to be included for scientific integrity. We take the point that the paper should be readable, so we have moved technical sections to the supplementary information and rewritten sections using simpler language where appropriate. We have taken on board many of the reviewer’s excellent suggestions and hope that the reviewer is happy with the compromise. The paper has been reduced from almost 6,000 words to just over 4,000 words and the discussion has been reduced to about a quarter of its original size.

It starts with the title that is not very catchy. I would suggest: “Molecular triggers of parosmia (drop the symptom of): novel insights based on gas chromatography analysis. I would not put underlying mechanisms in the title since the data do not support that.

We can adopt the new title, thank you for the nice suggestion.

However, we quite deliberately added “symptoms of”, as trigger in medical terms could mean the trigger of disease onset rather than the trigger of the symptoms. All three colleagues in various fields who read our paper prior to submission commented on this. We have therefore included the following caveat at Line 56. “Throughout the paper, when we refer to triggers, we are referring to triggers of the episodes rather than triggers of disease onset.”

Second: There is a mix of findings (line 81-182) and anecdotal findings (line 184-213) in individuals. This is weird and breaks the fluid reading. If you stick to this individual botanics, put all that into the supplement findings.

We have removed the Case studies from the paper.

Third: The authors lose too much time for enumerating all the potential mechanisms and try to fit their data into an explanation how parosmia are generated. The present data do not explain much about the mechanism.

See above, we have removed three quarters of this discussion.

They suggest a peripheral more than central elements involved but no proof is given with these results.

We refute this. The parosmic episodes are consistently and reliably triggered by a specific set of molecules at the periphery. These molecules do not pass into the brain and therefore act at the periphery. We believe this to be pretty strong evidence of a peripheral mechanism. However, we have never excluded a central mechanism (see line 307 in the revised MS), as there is also solid evidence for this too. We have added in line 329 “It of course does not exclude any contribution from a central mechanism, and indeed it is likely that both mechanisms are involved.”

Please shorten all that textbook section on parosmia mechanisms and focus on what you found!

We do believe that only experts in the field are aware of the textbook explanations and that definitions of the basic mechanisms are required to set the scenario for the non-specialists. The communication would be incomplete without them. We have reduced this slightly.

The real contribution you did is the following in my eyes. I would go for: Parosmia is frequent; Parosmia is difficult to measure and it is completely misunderstood how, hwy and why it occurs. Many mechanisms are put forward. We found a way to identify odor components that do reliably and in most patients elicit parosmia. This paper lays the potential ground for a) develop measurement tools in clinical routine to assess parosmia other than by questionnaires [1] or hedonic evaluations [2] and b) opens a potential way how to explore more scientifically the underlying mechanism of parosmia (and maybe phantosmia). That is the way I would sell the paper.

This is pretty much what we had intended, and hope that with the rewrite, these themes are now more obvious.

Thank you for these suggestions, we have added the following text at line 341 “Specific triggers could form the basis of such tests, providing better ways of measuring parosmia than questionnaires³⁵ or hedonic evaluations⁴¹. Furthermore, our work provides a potential tool to explore more scientifically the underlying mechanism of parosmia”

Finally, I have the impression that it is a little written in a too apodictic way. Like any neurological symptom, olfactory distortion (similar to phantom pain, tinnitus, etc..) may have several origins and not only one mechanism. Every neurological symptom may be due to central or peripheral pathologies or sometimes a conjunction of both. Please go for a discussion that considers that.

This was indeed our view that there are many origina, as we had already stated that our findings “neither prove nor refute the mis-wiring hypothesis” Line 317, and that a “a purely central causation seems unlikely” not dismissing, but rather incorporating the involvement of the central mechanism. For clarification we have added the following sentence at the end of the discussion in Line 330 “This of course does not exclude any contribution from a central mechanism, and indeed it is likely that both mechanisms are involved.” The word “suggest” has been added in line 291 and we have replaced “more likely to be” with “may be” in line 310, to make this less apodictic.

In line with that very black and white view it should also be stated that most patients with parosmia report coffee, cigarette smoke and some other odors as solid triggers, but that an enormous variety of parosmia trigger exist (e.g. even vanilla triggered parosmia has been described [3]).

Yes, we are aware that there are many other triggers of parosmia, and are currently identifying several more, but it is not necessary to find them all to demonstrate the principle. For clarification we have added the sentence in Line 281 “Undoubtedly, there are additional triggers in other foods, beverages, personal care products and even in the environment, but this subset found in coffee is sufficient to prove our hypothesis.”

I think the discussion merits mentioning that olfactory distortions are very varying.

I agree, thank you for pointing this out. This is now mentioned in Line 56 “and these distortions vary in strength and duration⁶” in Line 322 “the severity and duration of parosmia can fluctuate quite significantly on a daily basis⁶.”, and line 326 “why the response can fluctuate on a daily basis”

The present findings are encouraging that we may find ways to measure and “catch” the vast majority of parosmia.

Parosmia occurrence is not totally independent of olfactory status. Parosmia in anosmia patients is rare. But indeed, measured normosmia does not exclude parosmia. Please specify.

We have rephrased this in Line 184 “We demonstrate here that although on average most parosmic participants had a low olfactory function, parosmia also occurs in those with a normal olfactory

function and those who are functionally anosmic, and the relationship between quantitative and qualitative olfactory dysfunction is complex.”

References:

Please cite the position paper that gives a good overview on clinical smell disorders. It may help reduce over citing non relevant papers [4].

Apologies, we are very familiar with this excellent review and quote it a lot, but in this MS we have used others, mainly taking our figures from post-Covid data. Please indicate which references you find less relevant, and we are happy to replace them with [4].

Some other important papers on parosmia would be nice to be included. Reden et al [5] examined the occurrence of parosmia and phantosmia, showing that parosmia is largely related (much more than posttraumatic!!) to postinfectious events. As POID are associated more to purely peripheral events (local damage to the olfactory epithelium, in contrast to posttrauma) this may further help your discussion.

Added at Line 85 “The major inclusion criterion was those with post-infection olfactory loss, (the aetiology most likely to result on parosmia^{reden}) whilst those with degenerative olfactory loss, or loss due to traumatic brain injury were excluded from this study.” Also added at Line 328 “This is consistent with the local damage to the epithelium associated with post-infection olfactory loss”

Shorten the manuscript that will appear to those who do not want to read the supplements by 50 % and make it pleasant to read. Then it will be wonderful!

We feel that in reduction by 50% we may lose some of the scientific integrity but have reduced the manuscript by about a third and addressed the readability. The paper is much improved as a result and we hope that this is wonderful enough to merit publication!

1. Landis, B.N., et al., Evaluating the clinical usefulness of structured questions in parosmia assessment. Laryngoscope, 2010. 120(8): p. 1707-13.

Already quoted

2. Liu, D.T., et al., Assessment of odor hedonic perception: the Sniffin' sticks parosmia test (SSParoT). Sci Rep, 2020. 10(1): p. 18019.

Inserted at line 342

3. Frasnelli, J., et al., Clinical presentation of qualitative olfactory dysfunction. Eur Arch Otorhinolaryngol, 2004. 261(7): p. 411-5.

Inserted at line 207. Useful for difficulty in describing distortions, but don't need this for the ethyl vanillin reference.

4. Hummel, T., et al., Position paper on olfactory dysfunction. Rhinology, 2017. 54(Supplement 26): p. 1-30.

A very familiar paper, but we have not had the opportunity to cite it on this MS.

5. Reden, J., et al., A study on the prognostic significance of qualitative olfactory dysfunction. Eur Arch Otorhinolaryngol, 2007. 264(2): p. 139-44.

Inserted at line 85

Minor Comments:

I had to look up “tetrapartite symptom” since I did not understand this word. After extensive internet surfing I am still not smarter. Would it be possible to start your discussion with a word normal people understand?

It does exist, quadrapartite would be even better, but we take your point!

Reference 27: the journal is wrong: ORL and Related Spec, NOT J ORL HNS. Please correct.

Thank you for pointing this out, it has been corrected.

Reviewer #2 (Remarks to the Author):

Besides the well-known anosmia (loss of the ability to smell), the so-called parosmia is another rare disorder of the olfactory system. This disorder was little recognized in medicine, but with the COVID-19 pandemic it was observed that the virus infection caused anosmia in a certain percentage of the patients. Although most patients recovered after the infection, some developed a long-term dysfunction of the olfactory system, i.e. a parosmia. Such patients describe many food aromas as being intolerable and, thus, often normal food is rejected. In consequence they lose weight and may thus develop further clinical consequences.

The aroma of foods is caused by a mixture of several volatile components in the right concentrations, and today many food aromas have already been unraveled on a molecular basis by the application of the so-called sensomics concept. An important analytical tool of this concept is GC/olfactometry or the so-called GC/sniffing, which helps to identify the odor-active components in a food aroma distillate from the bulk of odorless compounds. In short, a panelist smells any volatile compound at a sniffing-port after separation of the entire volatile fraction by gas chromatography. This way, the aroma attribute of a single compound can easily be evaluated.

Parker and coworkers used the GC/O approach to detect the differences in aroma attributes evaluated by a group of participants with post-viral parosmia vs. those described by a group of participants without parosmia using the volatile fraction of a coffee beverage. By comparing the correct odor attributes given by the healthy subjects (using literature data on earlier identified coffee aroma compounds)

Please note that we had 15 normosmics complete the study too, and the data are our own – not literature

to the “wrong” attributes described by the parosmic panelists, differences appeared and such compounds were assigned by the authors as “molecular triggers of symptoms of parosmia”: I find the idea of using the GC/O approach very innovative, and the results are for sure very interesting in defining food aroma compounds responsible for the rejection of normal food by parosmic patients.

Thank you for these positive comments.

However, I cannot fully agree with the opinion of the authors with respect to the “underlying mechanism”.

The word mechanism has been removed from the title. Three quarters of the more mechanistic aspects have been removed, but we still feel we need to discuss our findings in light of the existing literature. See also responses to Reviewer 1 where we have clarified that both peripheral and central mechanism may be at play.

Also, sentences like “Furthermore, it provides the basis for development of a practical diagnostic tool and treatment strategies” (abstract, last sentence) are not supported by analytical data. Therefore, I suggest to tone down many sentences and to avoid speculations in the conclusions part.

Reviewer 1 recognises the potential for a diagnostic tool and we have expanded this on lines 345. As a result of this work, tests using known triggers are already being developed.

In treatment strategies we are referring more to dietary choice, meal planning and emotional support, rather than the development of therapeutic agents and we have clarified this in lines 344 onwards.

We have toned down some of our expressions (see response to Reviewer 1)

In addition, I miss systematic sensory experiments with pure reference compounds in defined concentrations, for example in aqueous solution. This way, the effect of the concentration of a compound on the olfactory epithelium, i.e. an olfactory receptor cell can correctly be studied.

We did use some pure compounds to confirm the identity of the triggers (section starting at Line 154), however, our rationale for NOT using reference compounds for the major body of work is given in the section starting at Line 104. Using a coffee extract allowed us to screen, in a much neater and more innovative way, many more aroma compounds with each participant than we could ever have done using single compounds. Having now identified some trigger compounds, we agree that measuring odour thresholds of single compounds could provide useful information and will undoubtedly form part of our further work. However, this is not necessary to prove our hypothesis.

Also, aroma profiles of the entire coffee beverage should have been performed to elucidate how the parosmic patients evaluate the beverage itself.

Firstly, consider the difficulties in carrying out aroma profiles with parosmic volunteers who usually can't start to describe what they are smelling in terms other than those that convey their disgust. There is typically one over-powering sensation of disgust.

Secondly, the key question we were addressing was not “Why does coffee cause parosmia?” but rather “Do highly potent aroma compounds, which happen to be present in coffee, trigger distortions?” As such, the coffee was used as a carrier of a wide range of trigger and non-trigger compounds (see section at line 104). We had ascertained from the screening questionnaire that coffee was producing distortions for every participant admitted to the study. We did not record their response to the Nescafe sachets, but feel that this is not necessary to prove our hypothesis.

Furthermore, to draw the right conclusions on the structure of compounds triggering parosmia, at least a few studies on positional isomers should have been done.

This has been carried out thoroughly and rigorously – please refer to the experimental details in the supplementary information, particularly Table S3, where we explain how we have assessed both analytically and organoleptically most of the compounds against authentic compounds on 2 columns – the standard approach for GC-O identification. We have no doubt over the structures of the stereoisomers.

Therefore, as they stand, although quite interesting, the data are a mainly descriptive.

Indeed, it is a qualitative study, but it is sufficient to prove our hypothesis.

Consequently the manuscript should be shortened considerably.

Manuscript has been reduced by about a third.

In addition, the use of the coffee beverage and the application of GC/olfactometry should be mentioned in the title.

We prefer to use the title provided by reviewer 1. We see the coffee as a carrier of potent odour compounds with which to demonstrate our point. The MS is not about coffee per se.

REVIEWERS' COMMENTS:

Reviewer #1 (Remarks to the Author):

The manuscript has been very extensively and nicely reviewed. I do not have any further comments.

Just one single minor remark: I would drop the milestone sentence in the discussion or change this word. The familiar readers will immediately notice the milestone character of the paper and for the other ones, this "milestone" self assessment will give the impression of authors with little modesty.